# Electroacupuncture Reduces Fibromyalgia Pain via Neuronal/Microglial Inactivation and Toll-like Receptor 4 in the Mouse Brain: Precise Interpretation of Chemogenetics

**DOI:** 10.3390/biomedicines12020387

**Published:** 2024-02-07

**Authors:** Sheng-Ta Tsai, Chia-Chun Yang, Hsien-Yin Liao, Yi-Wen Lin

**Affiliations:** 1Department of Neurology, China Medical University Hospital, Taichung 404332, Taiwan; tshengdar@gmail.com; 2School of Medicine, China Medical University, Taichung 404328, Taiwan; 3Department of General Psychiatry, Taoyuan Psychiatric Center, Ministry of Health and Welfare, Taoyuan 330035, Taiwan; peishiuan1010@gmail.com; 4School of Post-Baccalaureate Chinese Medicine, College of Chinese Medicine, China Medical University, Taichung 404328, Taiwan; 5Department of Acupuncture, China Medical University Hospital, Taichung 404328, Taiwan; 6Graduate Institute of Acupuncture Science, College of Chinese Medicine, China Medical University, Taichung 404328, Taiwan; 7Chinese Medicine Research Center, China Medical University, Taichung 404328, Taiwan

**Keywords:** electroacupuncture, fibromyalgia, microglia, TLR4, thalamus, chemogenetics

## Abstract

Fibromyalgia (FM) is a complex, chronic, widespread pain syndrome that can cause significant health and economic burden. Emerging evidence has shown that neuroinflammation is an underlying pathological mechanism in FM. Toll-like receptors (TLRs) are key mediators of the immune system. TLR4 is expressed primarily in microglia and regulates downstream signaling pathways, such as MyD88/NF-κB and TRIF/IRF3. It remains unknown whether electroacupuncture (EA) has therapeutic benefit in attenuating FM pain and what role the TLR4 pathway may play in this effect. We compared EA with sham EA to eliminate the placebo effect due to acupuncture. We demonstrated that intermittent cold stress significantly induced an increase in mechanical and thermal FM pain in mice (mechanical: 2.48 ± 0.53 g; thermal: 5.64 ± 0.32 s). EA but not sham EA has an analgesic effect on FM mice. TLR4 and inflammatory mediator-related molecules were increased in the thalamus, medial prefrontal cortex, somatosensory cortex (SSC), and amygdala of FM mice, indicating neuroinflammation and microglial activation. These molecules were reduced by EA but not sham EA. Furthermore, a new chemogenetics method was used to precisely inhibit SSC activity that displayed an anti-nociceptive effect through the TLR4 pathway. Our results imply that the analgesic effect of EA is associated with TLR4 downregulation. We provide novel evidence that EA modulates the TLR4 signaling pathway, revealing potential therapeutic targets for FM pain.

## 1. Introduction

Fibromyalgia (FM) is the third most common cause of musculoskeletal pain, next to low back pain and osteoarthritis in terms of prevalence [1]. FM prevalence is around 2.7% globally when using the 2010 modified classification criteria of the American College of Rheumatology (ACR) [2]. The total health care costs are estimated to be three times higher for FM patients than in other populations [3]. Furthermore, FM can cause an inability to work in patients and productivity loss in society [4]. FM is a complex, chronic pain condition that primarily involves the musculoskeletal system [5]. FM pain control cannot be achieved by medication alone [6]. This unsatisfactory pharmacologic response may be related to the complex mechanism of FM pain [1]. Current evidence suggests that immune cells, glia, and neurons form an integrated network that coordinates immune responses and modulates the excitability of pain pathways and results in central sensitization [7]. A human functional imaging study demonstrated higher neuroinflammation in a FM patients compared to healthy controls across several brain areas (such as the primary somatosensory cortex (SSC) and medial frontal gyrus) [8]. In 2008, a Japanese team published a mouse model of FM [9]. They used intermittent cold stress (ICS) to induce abnormal pain, mechanical allodynia, and hyperalgesia that lasted for more than two weeks. This model exhibits a sex hormone-independent female predominance of chronic pain, which is similar to clinical FM patients. Our previous research showed aggravated neuroinflammation in this ICS FM model [10,11].

Toll-like receptors (TLRs) recognize microbial components and initiate signal transduction pathways to induce gene expression. These gene products control innate immune responses and the direct the development of antigen-specific acquired immunity [12]. TLRs can be classified based on their location in the cell. TLR1, TLR2, TLR4, TLR5, and TLR6 are located on the cell surface. TLR3, TLR7, TLR8, and TLR9 are found within endosomes [13]. TLR4 is a type I transmembrane protein that can be activated by bacterial lipopolysaccharide (LPS) and can induce the production of pro-inflammatory cytokines and interferons [14,15,16]. The TLR4 pathway is expressed primarily in microglia and is crucial for regulating inflammatory responses [17,18,19,20]. During inflammation in the CNS, the innate immune system, including microglia, is activated by TLR4. Activated TLR4 then triggers intracellular signaling in two distinct pathways, including MyD88/NF-kB and TRIF/IRF3 [15]. The TLR4 signaling pathway was reported to be involved in major depressive disorders [21] and peripheral neuropathic pain [22]. A loss-of-function mutation in the Tlr4 gene reliably reduced microglial activation and led to lower levels of inflammatory mediators [23]. Antagonism of TLR4 attenuated NF-κB pathways and thus decreased the inflammatory mediators secreted by astrocytes [24]. Furthermore, TLR4 plays a crucial role in central neuroinflammation, including IL-1β upregulation [25].

Acupuncture is a traditional Chinese medicine technique, representing an ancient physiological system that believes health to be the result of harmony among bodily functions as well as between the body and nature [26]. Acupuncture was first developed around 3000 years ago and has been shown to provide therapeutic effects in many diseases [27], especially pain management. Compared to conventional treatment, several potential advantages are associated with acupuncture, including its low cost, low number of complications [28], the ability to offer a personalized treatment program, and fewer adverse effects [29]. Clinically, two meta-analyses summarized the efficacy of acupuncture in relieving stiffness and pain in FM patients [30,31]. Regarding its mechanism, our previous article showed that electroacupuncture reduced cold stress pain through downregulation of interleukins, TNF-α, and IFN-γ in mouse plasma [11]. Electroacupuncture reduced mechanical and thermal hyperalgesia in the same ICS mouse model of FM by attenuating brain TRPV1 signaling [10].

Chemogenetics is defined as a method whereby proteins are engineered to interact with previously unrecognized small molecules [32]. Among the various classes of chemogenetically engineered proteins, designer receptors exclusively activated by designer drugs (DREADDs) have emerged as the most widely utilized to date [1]. The pioneering research on G protein-coupled receptors by Bryan et al., published in *PNAS* [33], has garnered over 2000 citations to date. DREADD technology provides a minimally invasive approach to reversibly and remotely regulate the activity of selected brain areas through the systemic delivery of DREADD ligands [34]. Muscarinic receptor DREADDs, notably activated by clozapine N-oxide (CNO), are extensively employed, with the modified human M3 muscarinic receptor (hM3Dq) enhancing neuronal activity, while a modified M4 receptor (hM4Di) suppresses it [34,35]. In our experiment, we employed hM4Di to silence the SSC, aiming to mitigate central sensitization of pain in mice.

In this study, we aimed to determine the role of inflammatory mediators and the TLR4 signaling pathway in an ICS-induced FM pain mouse model. We hypothesized that inflammation underlies FM in mice through the activation of receptors such as TLR4. Furthermore, the therapeutic effect and detailed mechanisms of EA in this model are still unknown. We tested whether EA can relieve FM by regulating inflammatory mediators and related mechanisms. We also aimed to provide new evidence for the roles of EA and TLR4 in central sensitization. Chemogenetics precisely block SSC activity and display an analgesic effect through the TLR4 pathway.

## 2. Materials and Methods

### 2.1. Animals

In total, 40 female C57BL/6 mice, aged 8–12 weeks, were used in this study. After arriving, the mice were kept in a 12 h light-dark cycle with food and water ad libitum. A sample size of ten animals per group was calculated as the number required for an alpha of 0.05 and a power of 80%. In addition, the number of animals used here and their suffering were minimized. The laboratory workers were blind to treatment allocation during the experiments and analysis. The use of these animals was approved by the Institute of Animal Care and Use Committee of China Medical University (Permit no. CMUIACUC-2021-317), Taiwan, following the Guide for the use of Laboratory Animals (National Academy Press, Washington, DC, USA). Mice were subdivided into four groups: normal group (Group 1: Normal); ICS-induced FM group (Group 2: FM); 2 Hz electroacupuncture group (Group 3: 2 Hz EA), and Sham EA group (Group 4: Sham EA).

### 2.2. FM Model and Bio-Plex ELISA

All mice were maintained at room temperature, 24 ± 1 °C, before experiments. In the ICS-induced FM model [9], but not in the normal group, 2 mice were caged in a plexiglass cage (13 × 18.8 × 29.5 cm) covered with stainless-steel mesh. On the first day (day 0), the mice were kept in a cold room at 4 °C overnight (from 4:00 pm–10 am). The mice were next moved to 24 °C for 30 min at 10 am. After 30 min, mice were then moved back to the cold room at 4 °C for 30 min. This process was repeated until 4:00 pm. The mice were then placed in the 4 °C cold room overnight. Normal mice were kept at room temperature from days 0 to 7 of the experiment, with no interventions applied. Mice plasma was collected and analyzed using Bio-Plex cytokine assays (BIO-RAD, Hercules, CA, USA) [6].

### 2.3. EA Treatments

The mice were anaesthetized with 5% isoflurane for induction and then maintained in 1% isoflurane. Under anesthesia, a pair of stainless-steel acupuncture needles (1″ inch, 36G, YU KUANG, Tainan City, Taiwan) were bilaterally inserted at a depth of 3–4 mm into the murine equivalent of the human ZuSanLi (ST36) acupoints. The murine ST36 is located approximately 4 mm below and 1–2 mm lateral to the midpoint of the knee in mice. A bilateral subcutaneous transverse insertion of acupuncture needles into the scapular region was considered to be the sham acupoint [10]. In both the EA and sham EA groups, electrical stimuli were delivered using a Trio 300 stimulator (Ito, Japan) at an intensity of 1 mA for 20 min at 2 Hz with a pulse width of 100 μs. EA and sham EA treatment caused slight visible muscle twitching around the area of insertion. EA and sham EA stimulation was applied thrice from day 5 to 7, following the FM protocol.

### 2.4. Pain Behavior Test

The mechanical and thermal pain behaviors were determined 3 times from days 5 to 7 throughout the experiment after the induction of the FM model. All mice were moved to the behavior analysis room and were adapted to the environment for at least 30 min before behavior tests. All experiments were performed at room temperature, and the stimuli were applied only when the animals were calm but not sleeping or grooming. First, the von Frey filament test was conducted. Mechanical sensitivity was measured by testing the force of responses to stimulation with 3 applications of the electronic, calibrated von Frey filament (IITC Life Science Inc., Woodland Hills, CA, USA). Mice were placed onto a metal mesh (75 × 25 × 45 cm) and covered with a plexiglass cage (10 × 6 × 11 cm). Subjects were then mechanically stimulated using the tip of the filament at the plantar region of the right hind paw. The filament gram counts were recorded when the stimulation caused the subject to withdraw its hind paw. Second, the Hargreaves’ assessment was used to measure thermal pain behavior by testing the time of response to thermal stimulation with 3 applications using a Hargreaves’ test IITC analgesiometer (IITC Life Sciences, SERIES8, Model 390G). The mice were placed in a plexiglass cage on top of a glass sheet. The thermal stimulator was positioned under the glass sheet, and the focus of the projection bulb was aimed exactly at the middle of the plantar surface of the right hind paw. A cut-off time of 20 s was set to prevent tissue damage. In the thermal paw withdrawal test, the nociception threshold was assessed using the latency of paw withdrawal upon stimulus, and results were recorded when the constant applied heat stimulation caused the subject to withdraw its hind paw [6].

### 2.5. Western Blot Analysis

The mice were anaesthetized with 1% isoflurane and cervical dislocation. The thalamus, medial frontal cortex (mPFC), SSC, and amygdala tissues were immediately excised to extract proteins. Tissues were initially placed on ice and later stored at −80 °C, pending protein extraction. Total proteins were homogenized in cold radioimmunoprecipitation (RIPA) lysis buffer containing 50 mM Tris-HCl pH 7.4, 250 mM NaCl, 1% NP-40, 5 mM EDTA, 50 mM NaF, 1 mM Na_3_VO_4_, 0.02% NaN_3_, and 1× protease inhibitor cocktail (AMRESCO, Solon, OH, USA). The extracted proteins were subjected to 8% SDS-Tris glycine gel electrophoresis and transferred to a polyvinylidene difluoride (PVDF) membrane. The membrane was blocked with 5% non-fat milk in TBS-T buffer (10 mM Tris pH 7.5, 100 mM NaCl, 0.1% Tween 20). The membrane was then incubated with one of the following primary antibodies in TBS-T with 1% bovine serum albumin (BSA) for 1 h at room temperature: TLR4 (~35 kDa, 1:1000, Millipore, Burlington, MA, USA), MyD88 (~35 kDa, 1:1000, Millipore, USA), TRAF6 (~87 kDa, 1:1000, Millipore, USA), pERK1/2 (~42–44 kDa, 1:1000, Millipore, USA), pp38 (~41 kDa, 1:1000, Millipore, USA), pJNK (~42 kDa, 1:1000, Millipore, USA), and pNFκB (~65 kDa, 1:1000, Millipore, USA). Peroxidase-conjugated anti-rabbit antibody, anti-mouse antibody, or anti-goat antibody (1:5000) was used as the appropriate secondary antibody. The bands were visualized using an enhanced chemiluminescent substrate kit (PIERCE, Waltham, MA, USA) with LAS-3000 Fujifilm (Fuji Photo Film Co., Ltd., Tokyo, Japan). Where applicable, the image intensities of specific bands were quantified with NIH ImageJ 1.54h software (Bethesda, MD, USA) [6].

### 2.6. Immunofluorescence

Mice were euthanized using 5% isoflurane via inhalation and intracardially perfused with normal saline followed by 4% paraformaldehyde. The brain was immediately dissected and post fixed with 4% paraformaldehyde at 4 °C for 3 days. The tissues were placed in 30% sucrose for cryoprotection overnight at 4 °C. The brain was embedded in optimal cutting temperature (OCT) compound and rapidly frozen using liquid nitrogen before storing the tissues at −80 °C. Frozen segments were cut at 20 μm width on a cryostat and then instantaneously placed on glass slides. The samples were fixed with 4% paraformaldehyde and then incubated with a blocking solution consisting of 3% BSA, 0.1% Triton X-100, and 0.02% sodium azide for 1 h at room temperature. After blocking, the samples were incubated with primary antibodies (1:200, Alomone, Jerusalem, Israel), including TLR4 and Iba1, prepared in 1% bovine serum albumin solution at 4 °C overnight. The samples were then incubated with secondary antibodies (1:500), including 488-conjugated AffiniPure donkey anti-rabbit IgG (H + L), 594-conjugated AffiniPure donkey anti-goat IgG (H + L), and peroxidase-conjugated AffiniPure donkey anti-mouse IgG (H + L), for 2 h at room temperature before being fixed with cover slips for immunofluorescence visualization. The samples were observed using an epi-fluorescent microscope (Olympus, BX-51, Tokyo, Japan) with 20 × numerical aperture (NA = 1.4) objective. The images were analyzed using NIH ImageJ software (Bethesda, MD, USA) [6].

### 2.7. Chemogenetics Operation

All mice were anesthetized with isoflurane and fixed in a stereotaxic device. Then, a 23-gauge, 2 mm stainless cannula was inserted into the SSC, 0.5 mm posterior and 1.5 mm lateral of the bregma at 175 μm below the cortical superficial and fixed to the skull with dental glue. The inoculation cannula was inserted and connected with the Hamilton needle through a PE duct to inject 0.3 μL of viral solution more than 3 min through the pump (KD Scientific, Holliston, MA, USA). After injection, the cannula was well-maintained at SSC for additional 2 min to permit fluid to diffuse. Next, 0.3 μL of hM4iD DREADD (designer receptors exclusively activated by designer drugs: AAV8-hSyn-hM4D(Gi)-mCherry; Addgene Plasmid #50477) were injected into the SSC over two weeks. Clozapine N-oxide (CNO; Sigma, St. Louis, MO, USA, C0832) was injected to active the DREADD. CNO was thawed in 5% dimethyl sulfoxide (DMSO; Sigma D2650) and diluted with normal saline before intraperitoneal injection of 1 mg/kg at day 0 [35].

### 2.8. Statistical Analysis

Statistical analysis was performed using the SPSS 21 statistic program. All statistic data are presented as the mean ± standard error (SEM). Statistical significance among all groups was tested using the one-way ANOVA test, followed by a post hoc Tukey’s test. Values of *p* < 0.05 were considered statistically significant.

## 3. Results

### 3.1. Electroacupuncture Inhibits Cold Stress-Induced FM Pain in Mice

First, we examined the effects of EA in a cold stress-induced FM pain model to address the function of EA in the subacute phase. Before FM induction, all mice had similar mechanical responses that showed a normal distribution and no statistical significance between each group. Intermediate cold stress induced typical mechanical hyperalgesia (Figure 1A, D4: 2.48 ± 0.53, *n* = 10), as determined by the von Frey test, and was substantially attenuated by EA manipulation but not sham EA (Figure 1A, D7: 4.89 ± 0.46 and 2.74 ± 0.51, respectively). Next, we examined whether EA or sham EA would also alter thermal hyperalgesia in FM mice. The Hargraves’ test revealed significant thermal hyperalgesia in paw withdrawal latency after cold stress induction (Figure 1B, D4: 5.64 ± 0.32, *n* = 9). The latency decrease was reversed by EA but not sham EA (Figure 1B, D7: 10.21 ± 0.74 and 6.18 ± 0.69, respectively).

### 3.2. Inflammatory Mediators Were Increased in the FM Model and Reduced by EA but Not Sham EA

To test the role of inflammatory mediators in FM mice, we used a Bio-Plex ELISA technique to detect these molecules in mouse plasma. Before FM induction, inflammatory mediator levels were low in normal mice. Furthermore, IL-1β, IL-2, IL-5, IL-6, IL-9, IL-12, IL-17A, TNF-α, IFN-γ, and MCP-1 were increased in FM mice (Figure 2, * *p* < 0.05, *n* = 6). EA but not sham EA dramatically reversed these phenomena (Figure 2, ^#^
*p* < 0.05, *n* = 6).

### 3.3. EA but Not Sham EA Reduced Fibromyalgia Pain through Microglia and TLR4 Signaling Pathways in the Mouse Thalamus

We first used Western blot to measure ionized calcium-binding adaptor molecule 1 (Iba1), a microglia/macrophage-specific calcium-binding protein [2]. We found increased amounts of Iba1 in FM mouse brain (Figure 2A, black column, * *p* ˂ 0.05, *n* = 6). EA but not sham EA significantly attenuated this phenomenon (Figure 2A, black column, ^#^
*p* ˂ 0.05, *n* = 6). Furthermore, we explored TLR4 and its downstream effectors: MyD88, TRAF6, pERK, pp38, pJNK, and the transcriptional factor pNFκB. We found increased expression of TLR4 and downstream proteins in FM mice brain (Figure 2, * *p* ˂ 0.05, *n* = 6). Similarly, EA but not sham EA significantly attenuated this phenomenon (Figure 2, * *p* ˂ 0.05, *n* = 6). We performed immunofluorescence staining of the mouse thalamus. TLR4 and Iba1 were present at lower levels in normal mice but augmented in FM mice (Figure 2C, *n* = 4). EA but not sham EA meaningfully abridged these phenomena (Figure 2C, *n* = 4). Significantly increased double-positive signals were also observed in FM group, suggesting the colocolization of TLR4 and Iba1 (Figure 2C, *n* = 4). This pattern was abrogated by EA but not sham EA.

### 3.4. EA but Not Sham EA Reduced FM through Microglia and TLR4 Signaling Pathways in the Mouse Medial Prefrontal Cortex

We quantified the inflammatory mediators and TLR4 signaling pathway in the mouse medial prefrontal cortex. We found increased expression of Iba1- and TLR4-associated proteins in FM mouse brain: MyD88, TRAF6, pERK, pp38, pJNK, and pNFκB (Figure 3, * *p* ˂ 0.05, *n* = 6). EA but not sham EA significantly attenuated these patterns (Figure 3, * *p* ˂ 0.05, *n* = 6). TLR4 and Iba1 were present at minor levels in normal mice, but levels were amplified in FM mice (Figure 3C, *n* = 4). Management with EA but not sham EA significantly alleviated these findings (Figure 3C, *n* = 4). Meaningfully greater than before double-positive signals were also achieved in FM group, revealing colocolization of TLR4 and Iba1 (Figure 3C, *n* = 4). The phenomenon was abolished by EA but not sham EA.

### 3.5. EA but Not Sham EA Reduced FM through Microglia and TLR4 Signaling Pathways in the Mouse Somatosensory Cortex

We also found increased expression of proteins downstream of the Iba1 and TLR4 pathways in the somatosensory cortex (SSC) of the FM mouse model: MyD88, TRAF6, pERK, pp38, pJNK, and pNFκB (Figure 4, * *p* ˂ 0.05, *n* = 6). EA but not sham EA significantly attenuated this phenomenon (Figure 4, * *p* ˂ 0.05, *n* = 6). Immunofluorescence signals of TLR4 and Iba1 were low in normal mice but augmented in FM mice (Figure 4C, *n* = 4). EA but not sham EA management attenuated these signals (Figure 4C, *n* = 4). Increased dual signals were observed in the FM group (Figure 4C, *n* = 4). The increased signals were alleviated by EA but not sham EA.

### 3.6. EA but Not Sham EA Reduced FM through Microglia and TLR4 Signaling Pathways in the Mouse Amygdala

We found similar increases in expression in proteins downstream of the Iba1 and TLR4 pathways in the amygdala of FM mice: MyD88, TRAF6, pERK, pp38, pJNK, and pNFκB (Figure 5, * *p* ˂ 0.05, *n* = 6). EA but not sham EA significantly attenuated this phenomenon (Figure 5, * *p* ˂ 0.05, *n* = 6). Immunofluorescence results in amygdala indicated that TLR4 and Iba1 were low in normal mice and increased in FM mice (Figure 5C, *n* = 4). EA but not sham EA relieve these increases (Figure 5C, *n* = 4). Increased double-positive indications were observed in the FM group. These increases were abrogated by EA but not sham EA.

### 3.7. Chemogenetics Inhibition of SSC Attenuated Fibromyalgia Pain through the TLR4 Pathway

Figure 6A shows significant FM-induced mechanical hyperalgesia after induction (Figure 6A, black column, *n* = 6). Chemogenetics inhibition of the SSC meaningfully diminished mechanical hyperalgesia (Figure 6A, red column, *n* = 6). In thermal hyperalgesia, mice under FM induction presented noteworthy thermal hyperalgesia (Figure 6B, black column, *n* = 6). After CNO injection, thermal hyperalgesia was alleviated in mice under SSC inhibition (Figure 6B, red column, *n* = 6). Likewise, our results showed that TLR4 and associated molecules were reduced in the SSC of FM mice subjected to chemogenetics manipulation (Figure 6C, *n* = 6). Iba1 was not altered after CNO injection. TLR4, MyD88, TRAF6, pERK, pp38, pJNK, and pNFκB levels were reduced after CNO injection in the mouse SSC (Figure 6C, * *p* < 0.05, *n* = 6).

## 4. Discussion

We used the classical intermittent cold stress mice model to induce abnormal pain, mechanical allodynia, and hyperalgesia, with a gender hormone-independent female predominance of chronic pain, which is similar to that noted in clinical FM patients [9]. The behavior test confirmed the mechanical and thermal hyperalgesia in the FM mice. EA but not sham EA treatment reversed this hyperalgesia. Serum analysis showed increase pro-inflammatory cytokines in FM mice: IL-1β, IL-2, IL-5, IL-6, IL-9, IL-12, IL-17A, TNF-α, IFN-γ, and MCP-1. Similarly, EA but not sham EA treatment reduced these pro-inflammatory cytokines. Furthermore, molecular analysis found increased Iba1, TLR4, and related molecules in FM mice brain, indicating neuroinflammation. EA treatment reduced this neuroinflammation in FM mice. An illustration of our findings is presented in Figure 7.

Clinically, FM is complex and difficult to treat. It is known as “nociplastic pain” [36], a chronic pain that is maintained in part by central sensitization, a phenomenon of synaptic plasticity, and increased neuronal responsiveness in central pain pathways after stimulation [37]. FM has features of central sensitization [38], hyperalgesia, allodynia, and hypersensitivity to various external stimuli, such as sounds or lights [39,40,41]. A meta-analysis summarized 37 FM syndrome papers, including 1264 subjects, and found consistent results on differences between FM patients and controls in pain-related brain areas [39]. So, in our study, we focused on four mice brain areas: thalamus, prefrontal cortex, SSC, and amygdala. The thalamus and SSC are well-known pain-related brain regions. The prefrontal cortex receives projections from the mediodorsal nucleus of the thalamus [42]. The amygdala plays a crucial rule in the emotional processing of pain. The pain modulatory role of the amygdala is based on its projections to descending pain regions in the brain [43]. We found increase expression of microglia and the TLR4 pathway in these four brain regions.

Central sensitization is an important phenomenon and maintains the chronicity of pain. Accumulating evidence suggests that central sensitization is driven by neuroinflammation in the peripheral and central nervous systems [37]. A characteristic feature of neuroinflammation is the activation of glial cells, such as microglia and astrocytes, in the spinal cord and brain, leading to the release of proinflammatory cytokines and chemokines [44]. Currently some researchers use proton magnetic resonance spectroscopy (MRS) to evaluate brain neuroinflammation in patients with chronic pain [45]. Another study used an 18 kDa translocator protein (TSPO) as a marker of CNS inflammation and radioligand [11C]PBR28 bonded to TPSO. Then, they could visualize CNS inflammation using PET and MRI imaging. Most importantly, they found the signal of neuroinflammation decreased after local steroid injection, consistent with clinical pain release [46]. In our experiment, we used a chemogenetics method to localize the SSC and inhibit the SSC to reduce central sensitization and pain. Furthermore, a Japanese team created human-induced microglia-like (iMG) cells from human peripheral blood monocytes [47]. They found higher expression of tumor necrosis factor (TNF)-α at mRNA and protein levels in ATP-stimulated iMG cells from patients with FM compared to cells from healthy individuals. Interestingly, there was a moderate correlation between ATP-induced upregulation of TNF-α expression and clinical parameters of subjective pain and other mental manifestations of patients with FM. Their findings suggest that microglia in patients with FM are hypersensitive to ATP. In addition, TNF-α from microglia may be a key factor underlying the complex pathology of FM. According to the above research, we used the classical central sensitization mice model of FM to investigate neuroinflammation in the mouse brain, focusing on microglia.

Toll-like receptors (TLRs) are pivotal components of the innate immune response, shaping the adaptive immune response. However, TLR signaling pathways must be tightly regulated because undue TLR stimulation may disrupt the fine balance between pro- and anti-inflammatory responses [48]. TLR4 plays a role in chronic pain [49] and neuropathic pain [50], but is not specifically linked to FM. Our previous research on FM found alterations in TLR4 with electroacupuncture treatment [6].

Complementary and alternative medicine (CAM) is commonly used to treat patients with FM [51]. One of the most commonly used forms of CAM is acupuncture. For example, in Japan, the annual acupuncture utilization percentage ranges from 4.8% to 6.7%, with lifetime experiences of acupuncture ranging from 19.4% to 26.7% [52]. A recent meta-analysis summarized 15 neuroimaging studies of musculoskeletal pain, including FM, and revealed activated clusters in multiple cortical and subcortical brain structures in response to acupuncture, including the thalamus, insula, caudate, claustrum, and lentiform nucleus [52]. However, controlling the placebo effect is extremely important in acupuncture studies. In 1995, Beecher published the article “The powerful placebo” that started the discussion of the placebo effect [53]. In patients with FM, an interesting review including 124 studies (with 15,663 participants) found that placebo treatment affected pain and other patient-centered outcomes, such as sleep, fatigue, function, and overall quality of life [54]. Therefore, we compared true electroacupuncture and sham electroacupuncture in our study. Moreover, we selected the acupoint ST36 (Zusanli) because of its well-recognized analgesic effect in mouse pain models [55].

## 5. Conclusions

In conclusion, we used ICS to induce neuroinflammation and FM-like pain in mice. In this model, electroacupuncture reduced FM pain via neuronal/microglial inactivation and decreased TLR4 signaling in the mouse brain. This study opens a window of opportunity for the discovery and development of novel TLR4-targeted pain treatments in patients with FM. In this study, we analyzed the specific TLR4 pathway in the mice model of FM. Our data provide clinical implications of TLR4 in acupuncture analgesia. Future clinical trials should be performed to further investigate the role of TLR4 in acupuncture analgesia for FM.

## Figures and Tables

**Figure 1 biomedicines-12-00387-f001:**
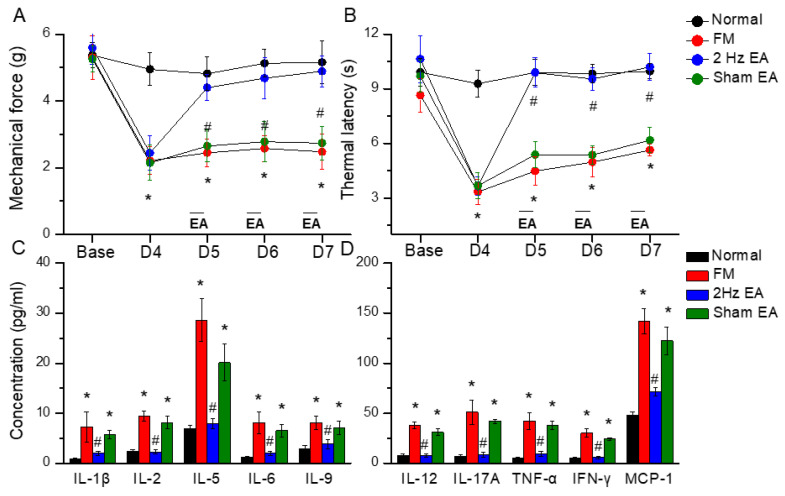
Mechanical withdrawal, thermal latency, and the concentration of inflammatory mediators in normal, FM, EA, and sham EA mice. (**A**) Mechanical threshold from the von Frey tests. (**B**) Thermal latency from the Hargreaves’ test. (**C**) IL-1β, IL-2, IL-5, IL-6, and IL-9 and (**D**) IL-12, IL-17A, TNF-α, IFN-γ, and MCP-1 in mice plasma. * means statistical difference when compared to the normal mice. ^#^ indicates statistical significance when compared to the FM groups. IL = Interleukin; IFN = Interferon; TNF = Tumor necrosis factor; MCP = Monocyte chemoattractant protein. *n* = 6 in all groups.

**Figure 2 biomedicines-12-00387-f002:**
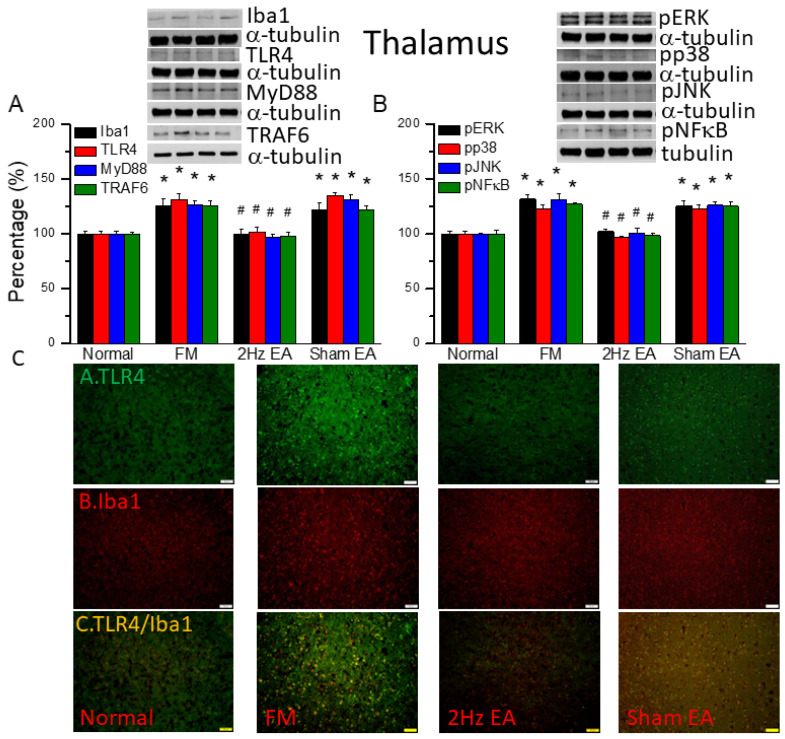
The levels of Iba1, TLR4, and related molecules in the mouse thalamus. Western blot bands contain four lanes of protein expression in the following order: normal, FM, 2 Hz EA, and sham EA. (**A**) Iba1, TLR4, MyD88, and TRAF6. (**B**) pERK, pp38, pJNK, and pNFκB. * indicates statistical significance when compared with the normal group. ^#^ indicates statistical significance when compared to the FM group. *n* = 6 in all groups. (**C**) Immunofluorescence staining of TLR4, Iba1, and double staining to assess protein expression in the mouse thalamus. Scale bar represents 100 μm. *n* = 4 in all groups.

**Figure 3 biomedicines-12-00387-f003:**
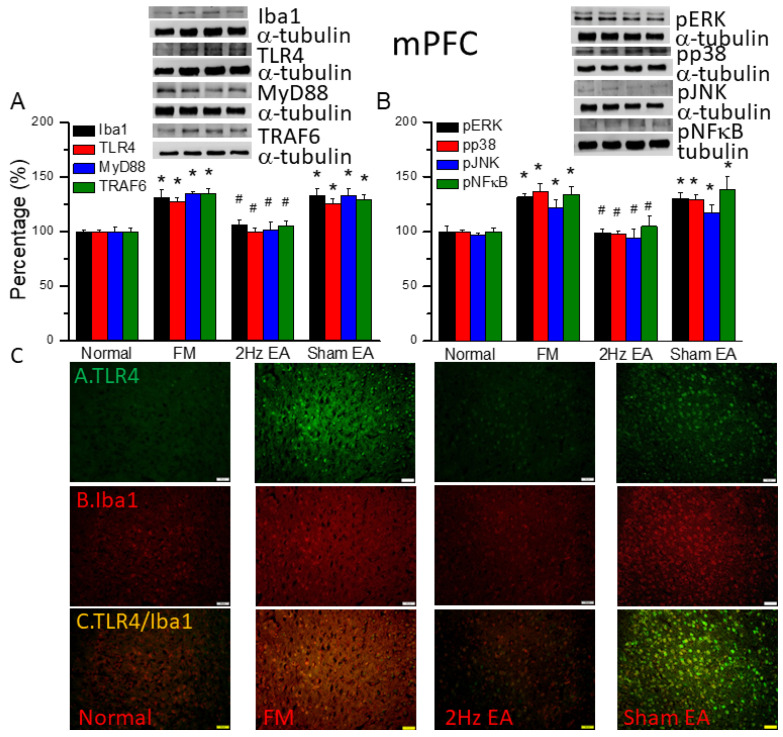
The levels of Iba1, TLR4, and related molecules in the mouse mPFC. Western blot bands contain four lanes of protein expression in the following order: normal, FM, 2 Hz EA, and sham EA. (**A**) Iba1, TLR4, MyD88, and TRAF6. (**B**) pERK, pp38, pJNK, and pNFκB. * indicates statistical significance when compared with the normal group. ^#^ indicates statistical significance when compared to the FM group. *n* = 6 in all groups. (**C**) Immunofluorescence staining of TLR4, Iba1, and double staining to assess protein expression in the mouse mPFC. Scale bar represents 100 μm. *n* = 4 in all groups.

**Figure 4 biomedicines-12-00387-f004:**
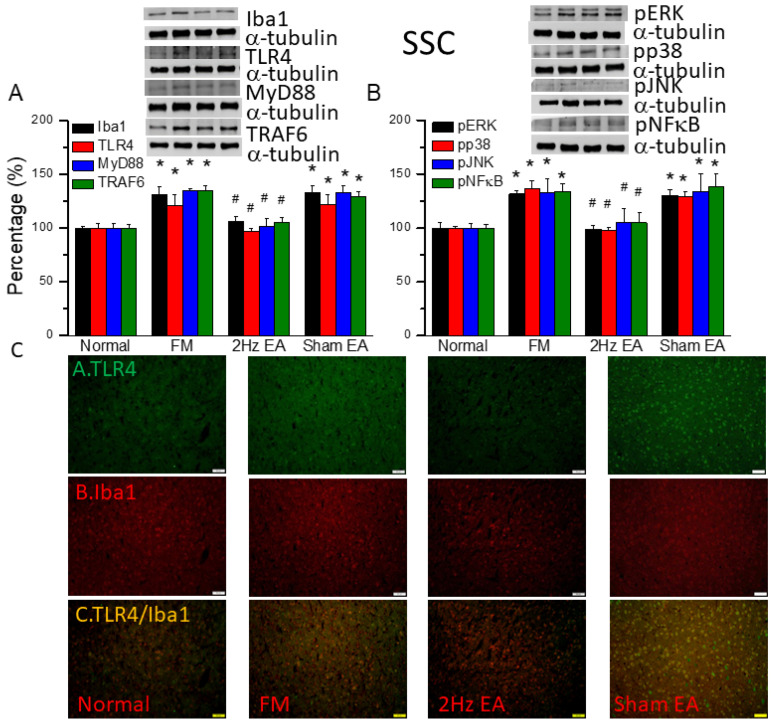
The levels of Iba1, TLR4, and related molecules in the mouse SSC. Western blot bands contain four lanes of protein expression in the following order: normal, FM, 2 Hz EA, and sham EA. (**A**) Iba1, TLR4, MyD88, and TRAF6. (**B**) pERK, pp38, pJNK, and pNFκB. * indicates statistical significance when compared with the normal group. ^#^ indicates statistical significance when compared to the FM group. *n* = 6 in all groups. (**C**) Immunofluorescence staining of TLR4, Iba1, and double staining to assess protein expression in the mouse SSC. Scale bar represents 100 μm. *n* = 4 in all groups.

**Figure 5 biomedicines-12-00387-f005:**
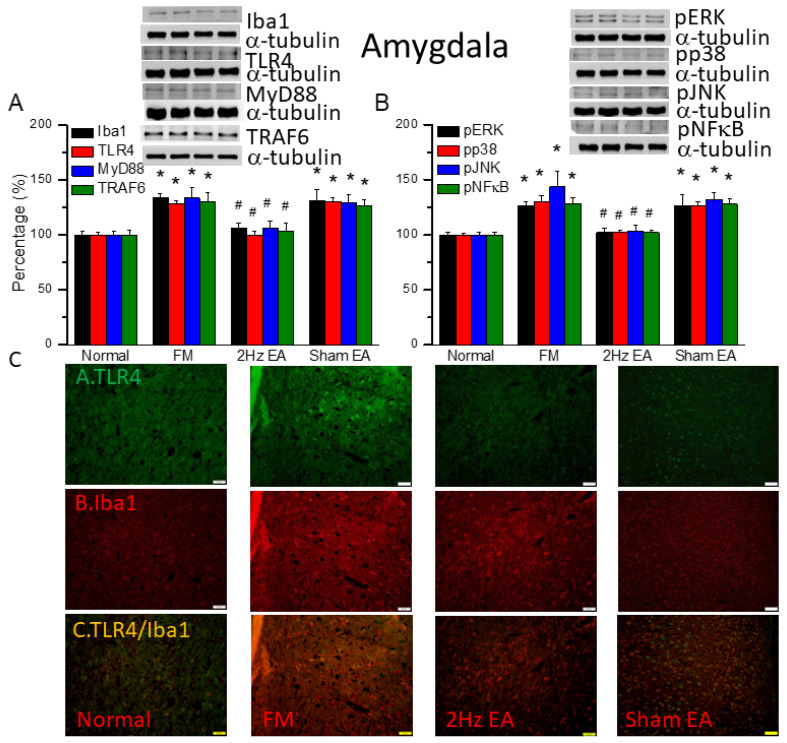
The levels of Iba1, TLR4, and related molecules in the mouse amygdala. Western blot bands contain four lanes of protein expression in the following order: normal, FM, 2 Hz EA, and sham EA. (**A**) Iba1, TLR4, MyD88, and TRAF6. (**B**) pERK, pp38, pJNK, and pNFκB. * indicates statistical significance when compared with the normal group. ^#^ indicates statistical significance when compared to the FM group. *n* = 6 in all groups. (**C**) Immunofluorescence staining of TLR4, Iba1, and double staining for protein expression in the mouse amygdala. Scale bar represents 100 μm. *n* = 4 in all groups.

**Figure 6 biomedicines-12-00387-f006:**
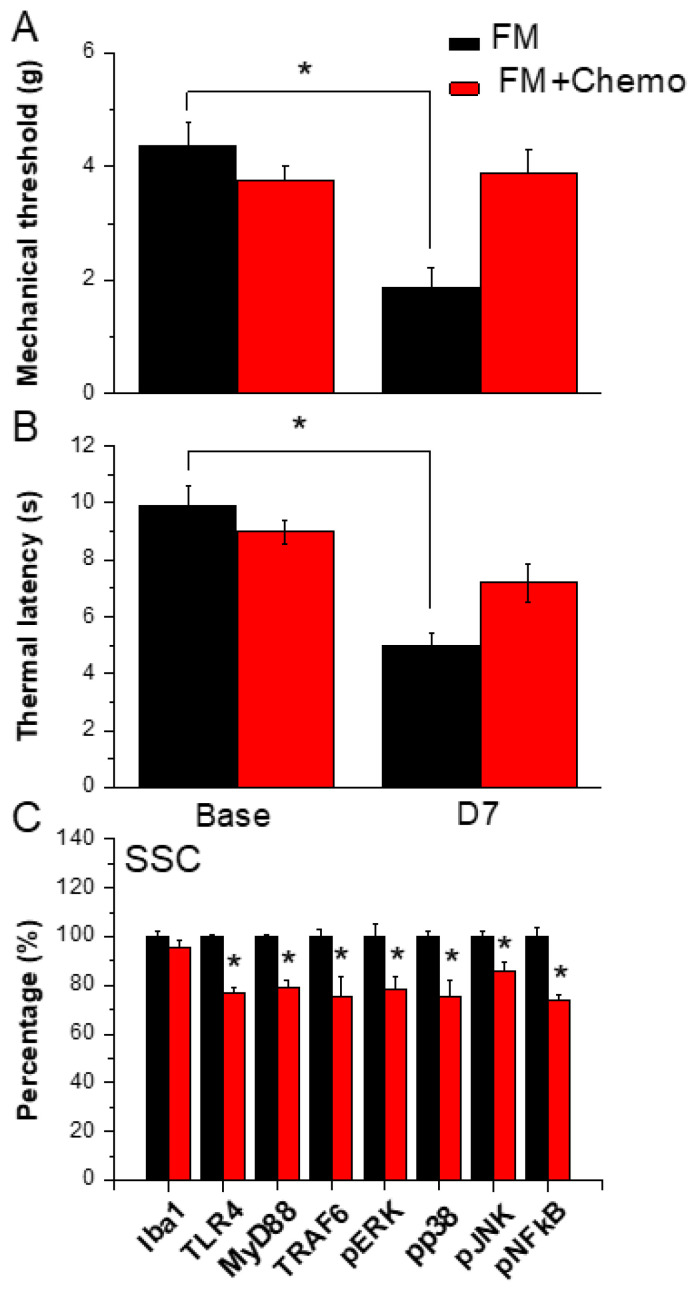
Pain behavior of FM and FM mice treated with the chemogenetics technique. Black column: FM group, red column: FM treated with chemogenetics. * *p* < 0.05 compared with the FM group. (**A**) Mechanical hyperalgesia (von Frey test). (**B**) Thermal hyperalgesia (Hargreaves’ test). (**C**) Protein levels of Iba1, TLR4, MyD88, TRAF6, pERK, pJNK, pp38, and pNFkB were measured in the mouse SSC.

**Figure 7 biomedicines-12-00387-f007:**
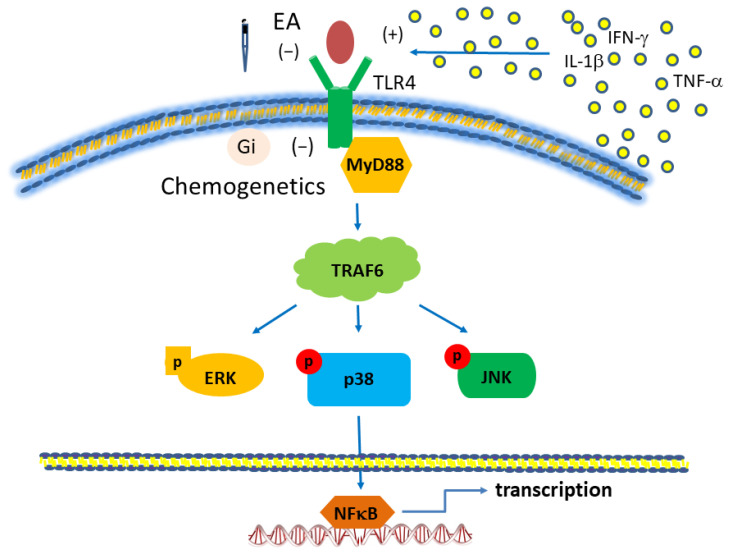
Toll-like receptor (TLR) signaling pathway with a focus on TLR4. Abbreviations: ERK = Extracellular signal-regulated kinase; IFN = Interferon; JNK = c-Jun N-terminal kinase; MyD88 = Myeloid differentiation primary-response protein 88; TLR, Toll-like receptors; TRAF = Tumor necrosis factor receptor-associated factor.

## Data Availability

The datasets supporting the conclusions of this article are included within the article.

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
