# Peer review of "Electroacupuncture Reduces Fibromyalgia Pain via Neuronal/Microglial Inactivation and Toll-like Receptor 4 in the Mouse Brain: Precise Interpretation of Chemogenetics"

_biomedicines, 2024, doi:10.3390/biomedicines12020387_

Round 1
Reviewer 1 Report
Comments and Suggestions for Authors
The article is interesting and well written. The analysis of the state of the art with respect to fibromyalgia and the underlying etiological mechanisms is in-depth and well presented, the same with regard to the presentation of the assumptions underlying acupuncture. the research design is appropriate and well explained. The results are well presented, also thanks to the support of images and their discussion is in line with them. Language is correct.
Author Response
Reviewer 1#
- The article is interesting and well written. The analysis of the state of the art with respect to fibromyalgia and the underlying etiological mechanisms is in-depth and well presented, the same with regard to the presentation of the assumptions underlying acupuncture. the research design is appropriate and well explained. The results are well presented, also thanks to the support of images and their discussion is in line with them. Language is correct.
RESPONSE: Thank you for your encouraging message. We made some modifications to make it better.
Reviewer 2 Report
Comments and Suggestions for Authors
The paper investigates the therapeutic potential of electroacupuncture (EA) in alleviating fibromyalgia (FM) pain in a mouse model. The study explores the role of Toll-like receptor 4 (TLR4) and inflammatory mediators in neuroinflammation associated with FM. The authors induce FM in mice through intermittent cold stress (ICS) and compare the effects of EA with sham EA. The findings reveal that EA, but not sham EA, has an analgesic effect, reducing mechanical and thermal hyperalgesia. TLR4 and inflammatory mediators are upregulated in specific brain regions of FM mice, indicating neuroinflammation, and are significantly reduced by EA. Additionally, a chemogenetics method targeting somatosensory cortex (SSC) demonstrates anti-nociceptive effects through the TLR4 pathway. The paper concludes that EA modulates the TLR4 signaling pathway, presenting potential therapeutic targets for FM pain.
The study provides strong evidence supporting the analgesic effect of electroacupuncture in a mouse model of fibromyalgia, addressing both mechanical and thermal hyperalgesia. The paper contributes to understanding the role of TLR4 and inflammatory mediators in neuroinflammation associated with FM. EA is shown to significantly reduce these markers in specific brain regions. The use of a chemogenetics method targeting SSC to demonstrate anti-nociceptive effects through the TLR4 pathway adds novelty to the research, suggesting potential pathways for therapeutic interventions.
The paper lacks an extensive discussion of related work, potentially limiting the context and understanding of the significance of the findings. A more comprehensive review of existing literature would strengthen the paper. The details of the electroacupuncture procedure, including electrode placement, stimulation parameters, and verification of needle placement, are not thoroughly discussed, raising concerns about the reproducibility of the results. The study employs a relatively small sample size, which may affect the generalizability of the results. Larger cohorts could enhance the robustness of the findings.
The paper effectively highlights the novelty of its findings by demonstrating the modulation of the TLR4 pathway through electroacupuncture in the context of fibromyalgia. The potential therapeutic implications make the research highly significant. The paper exhibits a commendable level of technical depth, especially with the use of a chemogenetics approach to dissect the involvement of the TLR4 pathway. However, providing more details on the EA procedure and its verification would enhance technical clarity. A more detailed description of the experimental setup, including precise EA parameters and verification methods, would aid in the reproducibility of the study. Clear documentation of the chemogenetics procedure and controls is also necessary. While the paper introduces the background on fibromyalgia and its association with neuroinflammation, a more extensive discussion of related work, especially the potential of the use of AI, such as Cerquitelli, Tania, et al. "Machine Learning Empowered Computer Networks." Computer Networks (2023): 109807, could better position the study in the existing research landscape. The paper is generally well-written and organized, but some sections, particularly the methodology, could benefit from additional clarity and details to facilitate understanding.
In conclusion, the paper significantly contributes to the understanding of electroacupuncture's therapeutic potential in fibromyalgia by modulating the TLR4 pathway. Addressing the identified weaknesses would strengthen the overall impact and reliability of the research.
Comments on the Quality of English Languageextensive editing is needed.
Author Response
Reviewer 2#
- The paper lacks an extensive discussion of related work, potentially limiting the context and understanding of the significance of the findings. A more comprehensive review of existing literature would strengthen the paper. The details of the electroacupuncture procedure, including electrode placement, stimulation parameters, and verification of needle placement, are not thoroughly discussed, raising concerns about the reproducibility of the results. The study employs a relatively small sample size, which may affect the generalizability of the results. Larger cohorts could enhance the robustness of the findings.
RESPONSE: Thank you for your detail review. We added a whole paragraph of Chemogenetics in “Introductoin” (Page 2, Lines 85-96, highlighted in yellow) for more comprehensive information. In total, we added five references. The detail electroacupuncture method is written at Page 3, Lines 129-140, highlighted in yellow (for example, we verify the needle placement by slight visible muscle twitching around the area of insertion, that’s a reproduciple phenomenon among different operators). The thorough discussion of the electroacupuncture is written at Page 13, Lines 398-409, highlighted in yellow (for example, we selected the acupoint ST36 because of its well-recognized analgesic effect in mouse pain models). About the study sample size, we described at Page 3, Line 110-112, highlighted in yellow: a sample size of ten animals per group (total four groups) was calculated as the number required for an alpha of 0.05 and a power of 80%. In addition, the number of animals used here and their suffering were minimized.
- The paper effectively highlights the novelty of its findings by demonstrating the modulation of the TLR4 pathway through electroacupuncture in the context of fibromyalgia. The potential therapeutic implications make the research highly significant. The paper exhibits a commendable level of technical depth, especially with the use of a chemogenetics approach to dissect the involvement of the TLR4 pathway. However, providing more details on the EA procedure and its verification would enhance technical clarity. A more detailed description of the experimental setup, including precise EA parameters and verification methods, would aid in the reproducibility of the study. Clear documentation of the chemogenetics procedure and controls is also necessary. While the paper introduces the background on fibromyalgia and its association with neuroinflammation, a more extensive discussion of related work, especially the potential of the use of AI, such as Cerquitelli, Tania, et al. "Machine Learning Empowered Computer Networks." Computer Networks (2023): 109807, could better position the study in the existing research landscape. The paper is generally well-written and organized, but some sections, particularly the methodology, could benefit from additional clarity and details to facilitate understanding.
RESPONSE: Thank you for these kind suggestion. The chemogenetics procedure was documented (Page 5, Line 201-212, highlighted in yellow). We also agreed that AI could facilitate current research, but we couldn’t find a suitable location to add this reference in our study.
- In conclusion, the paper significantly contributes to the understanding of electroacupuncture's therapeutic potential in fibromyalgia by modulating the TLR4 pathway. Addressing the identified weaknesses would strengthen the overall impact and reliability of the research.
RESPONSE: Thanks for your detail review again. We added one sentence “Future clinical trials should be performed to further investigate the role of TLR4 in ac-upuncture analgesia for fibromyalgia” to address the identified weaknesses to strength the overall impact and reliability of the research in the revised manuscript.
Reviewer 3 Report
Comments and Suggestions for Authors
Dear Author's
I read your article with great interest. The study was well prepared methodologically and properly presented the research results. I have no critical comments. Possibly one - References - some items are over 10-15 years old, doesn't this require correction? (unless they are essential...)best regards for all Author's
Author Response
Reviewer 3#
- I read your article with great interest. The study was well prepared methodologically and properly presented the research results. I have no critical comments. Possibly one - References - some items are over 10-15 years old, doesn't this require correction? (unless they are essential...)
RESPONSE: That’s a great point, thank you. We rechecked all of the references and replace …J Neurochem 88, 844-856 (2004) to an updated one:... International Journal of Molecular Sciences. 2024;25(3):1771.
And we deleted two old references: [36] Desmeules, J.A., et al. Neurophysiologic evidence for a central sensitization in patients with fibromyalgia. Arthritis Rheum 48, 1420-1429 (2003), and [49] Pioro-Boisset, M., Esdaile, J.M. & Fitzcharles, M.A. Alternative medicine use in fibromyalgia syndrome. Arthritis Care Res 9, 13-17 (1996), as your suggestion.
Reviewer 4 Report
Comments and Suggestions for Authors
This study by Tsai et al, presents a comprehensive understanding of how electroacupuncture effectively mitigates fibromyalgia pain in mice by elucidating the intricate interplay of neuronal and microglial inactivation, coupled with the modulation of Toll-Like Receptor in the mouse brain, offering a precise interpretation through the lens of chemogenetics. The minor revision as per comments suggested herewith.
1. Line 84 , hypothesized that inflammation underlies FM in mice through the activation of receptors such as TLR4. This means that it was not proven or study previously?
2. Line 92: Our data provide novel evidence to support the clinical use of EA in treating FM. Avoid such claim at introduction section.
3. Cite all methods, properly.
4. Clarity of figures, is poor, improve it, specially font size in figures.
5. Line 325-327, We used the classical intermittent cold stress mice model to induce abnormal pain, 325 mechanical allodynia and hyperalgesia, with gender hormone-independent female pre- 326 dominance of chronic pain, which was similar to clinical fibromyalgia patients. Please cite this work
6. Line 387: In patients with fibromyalgia, an interesting review including 124 studies found that placebo treatment affected pain and other patient centred outcomes such as sleep, fatigue, function, and overall quality of life, is this explains about review or research?
7. Remove figure from conclusion. Please place it appropriately or remove it.
8. Author section, Sheng-Ta Tsai 1, 2, Chia-Chun Yang 3, Hsien-Yin Liao 4, 5, *, Yi-Wen Lin 6, 7 and *….it looks like incomplete ..or error.
Author Response
Reviewer 4#
- Line 84, hypothesized that inflammation underlies FM in mice through the activation of receptors such as TLR4. This means that it was not proven or study previously?
RESPONSE: Yes, TLR4 was not a widely investigated pathway in patients with fibromyalgia. Only some studies discussed about it (Page 2, Line 55-71). Inflammatory mediators such as S100B and HMGB1 may activate TLR4 then triggers intracellular signalling. This study suggests that several cytokines (IL-1b, IL-2, IL-5, IL-6, IL-9, IL-12, IL-17A, TNF-a, IFN-g, and MCP-1) may activate TLR4 for mice fibromyalgia induction.
- Line 92: Our data provide novel evidence to support the clinical use of EA in treating FM. Avoid such claim at introduction section.
RESPONSE: Thanks for your remind, we deleted the sentence to avoid such claim at introduction section.
- Cite all methods, properly.
RESPONSE: We rechecked the “Materials and Methods” section and cited some important papers.
- Clarity of figures, is poor, improve it, especially font size in figures.
RESPONSE: Thanks for your suggestion, we adjusted the words especially font size in figures in the figures.
- Line 325-327, We used the classical intermittent cold stress mice model to induce abnormal pain, 325 mechanical allodynia and hyperalgesia, with gender hormone-independent female pre- 326 dominance of chronic pain, which was similar to clinical fibromyalgia patients. Please cite this work.
RESPONSE: We cited this work in Page 11, Line 337.
- Line 387: In patients with fibromyalgia, an interesting review including 124 studies found that placebo treatment affected pain and other patient centred outcomes such as sleep, fatigue, function, and overall quality of life, is this explains about review or research?
RESPONSE: It explained the findings of this review, including 124 trials, total 15,663 participants. To make it clear, we added the number of the number of participants, at Page 13, Line 405.
- Remove figure from conclusion. Please place it appropriately or remove it.
RESPONSE: We replaced it after the first paragraph of “Discussion”. (Page 12)
- Author section, Sheng-Ta Tsai 1, 2, Chia-Chun Yang 3, Hsien-Yin Liao 4, 5, *, Yi-Wen Lin 6, 7 and *….it looks like incomplete ..or error.
RESPONSE: That’s an error, we corrected it. Thanks for your careful review.
Round 2
Reviewer 2 Report
Comments and Suggestions for Authors
All my comments have been addressed.